# A Reliable Merging Link Scheme Using Weighted Markov Chain Model in Vehicular Ad Hoc Networks

**DOI:** 10.3390/s22134861

**Published:** 2022-06-27

**Authors:** Siman Emmanuel, Ismail Fauzi Bin Isnin, Mohd. Murtadha Bin Mohamad

**Affiliations:** Faculty of Engineering, Universiti Teknologi Malaysia, Skudai 81310, Malaysia; ismailfauzi@utm.my (I.F.B.I.); murtadha@utm.my (M.M.B.M.)

**Keywords:** weighted Markov chain, clustering, weight value, merge window, merging link, merge collision, predicting probability

## Abstract

The vehicular ad hoc network (VANET) is a potential technology for intelligent transportation systems (ITS) that aims to improve safety by allowing vehicles to communicate quickly and reliably. The rates of merging collision and hidden terminal problems, as well as the problems of picking the best match cluster head (CH) in a merged cluster, may emerge when two or more clusters are merged in the design of a clustering and cluster management scheme. In this paper, we propose an enhanced cluster-based multi-access channel protocol (ECMA) for high-throughput and effective access channel transmissions while minimizing access delay and preventing collisions during cluster merging. We devised an aperiodic and acceptable merge cluster head selection (MCHS) algorithm for selecting the optimal merge cluster head (MCH) in centralized clusters where all nodes are one-hop nodes during the merging window. We also applied a weighted Markov chain mathematical model to improve accuracy while lowering ECMA channel data access transmission delay during the cluster merger window. We presented extensive simulation data to demonstrate the superiority of the suggested approach over existing state-of-the-arts. The implementation of a MCHS algorithm and a weight chain Markov model reveal that ECMA is distinct and more efficient by 64.20–69.49% in terms of average network throughput, end-to-end delay, and access transmission probability.

## 1. Introduction

In a vehicle ad hoc network, topological changes are frequent as nodes (vehicles) move in accordance with traffic laws [1,2,3]. As automobile density grows, access collisions occur as a result of poor packet data transmission during slot allocation [3]. Therefore, an effective clustering can lengthen the lifespan of a network. Clustering is a technique for dissecting a network’s architecture. Topological data are obtained more quickly due to the network’s smaller size (cluster). Because of the lack of centralized administration, network topology management and resource allocation become difficult, resulting in inefficient throughput and increased access latency [1]. To overcome hidden terminal problems and merge collisions, an effective clustering technique is required. When using an allocated technique to assign a period allocation [4], two sorts of conflicts can occur: access collision and merge collision. Due to mobility, two vehicles that started more than two hops apart try to join a single period allocation at the same moment [5]. Automobiles traveling in reverse directions with RSUs fastened to the road [6] and two or more clusters merging can cause merging conflicts. Assuming the nodes in this scenario have already been assigned time slots in their clusters during the cluster merging process, they must be released from their current time slot to acquire a new one, which may result in merging collision [7], whereas access collision occurs when more than one node (that has not yet acquired a time slot) within transmission coverage, or approximately two hops apart, attempts to enter a single available period allocation. Therefore, increasing traffic density that is not in cluster may cause hidden terminal issues and access conflicts, leading in inefficient medium usage and increasing access delays. When the IEEE 802.11p MAC detects an idle channel, it instantly initiates transmission or selects a backup value from the contention window (CW) and initiates a countdown phase [8]. When a vehicle transports a high quantity of data packets, it participates in multiple competitions. Because 802.11p does not support RTS/CTS for data packet broadcasting [9], it is susceptible to hidden terminal issues and conflict [10]. Therefore, collisions between data packets are not immediately noticeable. The TDMA protocol was proposed for automotive ad hoc networks in order to improve transmission efficiency and overcome IEEE 802.11p’s restrictions. Numerous shared TDMA-based MAC principles for VANETs have been presented that aimed to eliminate or mitigate merging conflicts as well as the hidden terminal problems [8,9]. In the centralized TDMA protocol, the centralized node allocates the time slot, and in the distributed TDMA protocol, each node manages the time slot [11,12,13,14]. Due to the high vehicle density, the TDMA Cluster MAC (TC-MAC) recently modified the approach for allocating TDMA slots in group-based (Cluster) VANETs. Unlike DSRC, TC-MAC maintains a better level of reliability for safety messages [15,16].

Due to the widespread use of VANETs, an intelligent transport system must transfer data to several nodes [17,18]. When vehicles are partitioned into virtual clusters, scalability of the network becomes a challenge. In [17], clusters are led by a cluster head (CH), which is assisted by cluster members (CM). When merging leads in bigger clusters [18], numerous recent clustering methods employ a small intra- and inter-cluster process size (Figure 1) [19]. The enhanced weight-based clustering algorithm (EWCA) [17] was demonstrated in a cluster. It considers the time and position of vehicles in the cluster, and it is assumed that vehicles are traveling at similar speeds. Every node within broadcast range of its nearest neighbors was considered. This was done to ensure cluster stability and the effective transmission of safety data. As a result, the techniques are more suited to a single traffic condition, and the mobility component is overlooked, resulting in access and merging crashes with all automobiles travelling at constant speed in a medium-density environment.

In [20], MAMC-MAC protocol was developed to increase VANET reliability and to convey safety alerts. It utilized a hopsector message direction schema to maximize the message delivery into a particular domain in real time. TDMA is used to divide the dedicated short range communication (DSRC) band into frames. The MAMC-MAC protocol is more appropriate for a single traffic scenario, and the mobility factor is not taken into account. Furthermore, all vehicles are traveling at a constant speed in a medium traffic density, and also the clusters merging scenario is not taken into account. It is likely to result in access collisions and merging collisions. In addition, there are hidden terminal issues, resulting in merging collisions. The cluster merging process is the process which merges two adjacent clusters into a larger cluster. This process occurs during a period called the merging window (Mw). Based on the above issues, our contributions are as follows:We studied and created a merge cluster head selection (MCHS) algorithm that minimizes the frequencies of merging collision and hidden terminal concerns while also electing the best-fit CH in a merged cluster when two or more clusters merged.We used a weighted Markov chain model to describe the transformation operation within a cluster and differentiated it from other clusters based on the weighted value.During the clusters merging window, the weighted Markov chain mathematical model improved accuracy while reducing ECMA channel data access transmission delay with unmatched transition speed in a time slot and state slot (frame). During the window time (T), this speeds up transitions while eliminating hidden terminal difficulties and access collision.The aperiodic MCH selection is based on the merge window period probability and the development of a centralized cluster in a VANET where all nodes are one-hop nodes.In a merged cluster, the CHs choose the best candidate as the MCH. Although their cluster members (CMs) inside the transmission range released their time slots and acquired a new time slot from the new MCH, the other CHs became CMs. The CMs that are outside of the new MCH’s transmission range will continue to cling to their previous CH, which has now turned to gateway node (Gw), until all of the remaining CMs are inside the MCH’s transmission range, at which point the Gw will eventually switch to CM.For performance evaluation, we built a detailed simulation model and put the suggested technique into action. Extensive simulation results indicated the superiority and scalability of the proposed ECMA method.

The remainder of this paper is organized as follows. Section 2 provides the existing works on merging collisions. Section 3 describes the proposed method that applies the new weighted Markov chain and the cluster head selection algorithm in the merged cluster. The fourth part presents the performance evaluation of the selected indicators, simulation parameters, and their values, followed by Section 5 and Section 6 as the Discussion and Conclusion respectively.

## 2. Existing Works on Merging Collisions

The set of vehicles, called a cluster, allows vehicles to communicate with their neighbors, called intra-cluster networks, and two or more cluster may communicate with each other, called inter-cluster networks [13,14]. When two clusters are in a merging process the node is relieved of their access time slot and acquires new ones which may result in merging collisions. Because merging collisions occur due to vehicle movement and are marginal based on time slot size [15,16,17], Vehicle ad-hoc network media access control protocol (VeMAC) calculates the rate of merging collisions in the time frame rather than in the time slot. Although access collision occurs when nodes attempt to acquire a time slot, a merging collision occurs when vehicles have successfully acquired a time slot. It can also occur when cars traveling in the same direction but at different speeds. As node *x* moves into THS2 (Two hopes state II) and shares the same time slot as node *z*, a merging collision occurs at *z*, as shown in Figure 2. As a result, when a node detects a merging collision on an access channel, it releases its time slot and acquires a new one, resulting in an access collision, especially with hidden terminal occurrence. In [21,22], a distributed algorithm was used in VeMAC, which requires two vehicle transceivers, one tuned to the control channel and the other to the service channel. However, because of the large size of the control frame in VeMAC, contact over the control channel becomes an overhead.

In [23], the Direction based clustering and multi-channel medium access control (DA-CMAC) protocol is an extension of the VeMAC protocol to improve the transmission reliability of the safety message, where RSU, GW, and CH consist of time slots for cluster members (CMs) that the work divide into two, depending on the location of the vehicles to reduce the rate of change in DA-CMAC access and merging collisions. In addition, each cluster member (CM) is given one slot in both the control and service channels to ensure channel access equity. The application of the Gw node [24] and the allocation of slots from the RSU to CMs in different directions at different speeds result in merging collisions [25] when two or more clusters merge. In [26], the authors used leadership-based clusters (LCM) merging to investigate the impact of merging collisions in a cluster vehicular ad hoc network. When clusters in the same direction combine to form a new unified cluster, the scheme assigns the best connection to each cluster member and remains stable. Two CHs moving in the same direction and in the same transmission range would activate the merging detection mechanism in this scheme. That study only examined the impact of hidden terminal problems and the transmission of access data in a merging cluster by relying on the identification mechanism for cluster stability mergers. In [27], researchers used a cluster merging mechanism in CCFM-MAC to avoid cluster merging until they were sufficiently close to each other. The Hello or cluster head packets are received by all the cluster heads within the transmission range between each. When the CHs are within a certain time interval of each other’s communication range, the CHs repeat the Hello and CH packet responses, and the clusters merge. When two or more clusters combine, the CH with the highest ALERT remains the CH, while the other cluster members are dissolved. The goal is to avoid cluster merging in a short period of time and to increase cluster stability. However, gateway vehicles are used as a connection sub-domain that allows CMs and CHs to link. Hidden terminal problems and collision-free clusters [7] are solved using this scheme. Disjointed time slot sets are associated with distinct lanes on the same road segment and distinct road segments at intersections, according to MoMAC [28]. Furthermore, each vehicle transmits safety messages along with time slots occupying neighboring vehicle data; vehicles can detect time-slot collisions and access a vacant time slot in a completely distributed manner by updating time slots occupying information from two-hop neighbors (obtained indirectly from one-hop neighbors) using Chain Markov [29,30]. In this situation, two CHs which are in the same contact range resolves to cluster merging [31]. The CH with the highest weight value [32], on the other hand, will continue to be the CH, while the others stepped down. CMs have the option of joining the new leadership, joining another cluster, or forming their own. In a cluster-based TDMA MAC protocol [33], the duty of the CH is to assign a slide of time to the CMs. Meanwhile, the authors have only looked at cluster stability and have yet to incorporate the dynamic slot allocation approach into a clustering mechanism to reduce the rate of merging collisions.

The extended delay problem that vehicles can encounter in the event of a merging collision with a TDMA-based MAC protocol for VANET is highlighted and formulated in [34]. This study demonstrated that this delay is proportional to the number of collided packets immediately following the merging collisions. The proposed slot suggestion system is used to prevent additional access collisions between vehicles that have vacated their time slots because of a slot-merge collision. After an access collision, this method minimizes the likelihood of additional access collisions [5].

## 3. Proposed Method

In this section, we proposed the Merge Cluster-Head Selection algorithm (MCHS) to reduce the rates of merging collisions and hidden terminal problems when two or more clusters merge, as well as the selection of the best match MCH in a merged cluster. Merging collisions occur when vehicles from separate clusters combine to enter a shared time slot. When the algorithm is used, as presented in Algorithm 1, the merging collision can be significantly reduced. By adapting clustering and MCHS algorithm in a merged cluster, the proposed ECMA protocol attempts to achieve collision-free in a cluster while also minimizing the rate of merging collisions in an inter-cluster VANET.

As the different clusters at ‘M’ converge, they come into contact as in ‘N’, where the two CHs are in the transmission range of each other. In ‘L’, the appropriate CH becomes the MCH, the other CH becomes Gw and remains attached to the CMs beyond the MCH’s transmission range, and in ‘Q’, all the cluster members (CMs) are inside the MCH’s transmission range, and the Gw becomes a CM, as shown in Figure 3.

### 3.1. Weighted Markov Chain

During the clusters merging window, the weighted Markov chain mathematical model enhances accuracy and minimizes ECMA channel data access transmission delay with unmatched transition speed in timeslot and state-slot (frame). This speeds up transitions while avoiding hidden terminal issues and access collision during the window time (T). In the frame’s frequency state of slot reservations, self-correlation coefficients represent various reservation prevalence packet data relationships. The frequency state of slot reservation in present frames can be used to predict the frame prevalence packet data in the future. Then, in comparison to the future frames, a weighted average based on the frequency of other current frames slot reservation can be calculated. As a result, the prediction goal of making full and equitable use of knowledge has been met. This is the fundamental idea behind weighted Markov chain prediction [35,36].

A branch of the Markov chain process is the weighted Markov chain [37]. If the system’s present state is given, then (conditionally) the past and future are independent. Such an action is referred to as the system’s Markov property. In a discrete (countable) state space with respect to discrete or continuous time, a Markov chain evolves.

A stochastic process *X* = {*X*(t),t∈T} is defined on a probability space (Ω, *F, P*), where parameters set T = {0, 1, 2, …}, and state space E = {0, 1, 2, …}.
$$P\{X\left(m+k\right)=i_{m+k}\;|\;X\left(m\right)=i_{m},X\left(j_{1}\right)=j_{1},\ldots ,$$
$$X\left(j^{2}\right)=j^{2},X\left(j_{1}\right)=j_{1}\}=P\{X\left(m+k\right)$$
(1)$$=i_{m+k}\;|\;X\left(m\right)=i_{m}\}$$

The general time-slot transition step is given by $P^{t}$ for any slot time t. The Markov chain nodes *X*_0_, *X*_1_,…, *X*_n_ have a slot time reservation state, S = (1, 2, 3, …, *n*), (Figure 4), where the Transition Matrix P element is defined as:(2)$$P\left(X_{t}=j|X_{0}=i\right)=P\left(X_{n+t}=j|X_{n}=i\right)={\left(P^{t}\right)}_{\mathrm{ij}}\;\text{for any n}.$$

The window period probability is $\pi_{j}$ (3), and the mean recurrence time to state j is $\mu_{jj}$. Taking the inverse of the mean recurrence time is one technique for determining the window period probability, as shown by the preceding identity.
(3)$$\pi_{j}=\frac{1}{\mu_{jj}}$$

An ergodic Markov chain is an irreducible Markov chain that is aperiodic and positive recurrent. Equation (4) depicts the ergodic chain’s finite distribution j, which is the only nonnegative solution to the equations.
(4)$$\{\begin{array}{c} \pi_{j}=\overset{\infty }{\underset{k=o}{\sum }}\pi_{k}\;p_{\mathrm{kj}}\;\;j=0,\;1,\;2,\ldots \\ \overset{\infty }{\underset{j=o}{\sum }}\pi_{j}=1 \end{array}$$

The Markov chain’s long-run proportion of time spent in state j can now be written as *π*_j_. Based on the above Markov chain and the window period probability, the specific method of weighted Markov chain prediction is expressed as follows [38]:

Determine a criterion for categorizing the frame’s incidence of slot reservation based on the length of the super frame and the specific adaptability requirement. The distance between the ideal one-hop node (OHN) and the CH in two-dimensional Euclidean space is expressed as E = 1, 2, 3, 4, and so on. The frequency condition of slot reservation is determined for each frame based on the classification standard of the threshold value (Sthr). Equation (5) is used to calculate the various self-correlation coefficients $r_{k}$, *k* ∈ ∆Sthr, where $r_{k}$ denotes the k-frame self-correlation coefficient, $x_{1}$ = 1, 2, …, *n* denotes the *i*th frame slot reservation prevalence, x denotes the mean value of $x_{1}$, and *n* denotes the length of the slot reservation series’ frame frequency state.
(5)$$r_{k}=\overset{n-k}{\underset{l=1}{\sum }}\left(x_{1}-\bar{x}\right)\left(x_{1+k}-\bar{x}\right)/\overset{n}{\underset{ı=1}{\sum }}{\left(x_{1}-\bar{x}\right)}^{2}$$

We create a diverse set of self-correlation coefficients and use them consistently. The weights of multiple (steps) Markov chains must also be considered (m is the maximum step predicted). As the prediction probability index, take the weighted average ${𝒲}_{k}$ of the various predicting probabilities for the same condition as shown in Equation (6).
(6)$${𝒲}_{k}=\left|\;r_{k}\;\right|/\overset{m}{\underset{k=1}{\sum }}\left|\;r_{k}\;\right|$$

We can derive various phases of Markov chain transition probability matrices from the statistical results from slot reservation prevalence transitions, which determine the probability rule. For example, in a different frame, the frequency of slot reservation $P_{i}\left(k\right)$, isthr can be predicted and combined with the relative transition probability matrices of a different frame, where k is the Markov chain step and k = 1, 2, …, m. If $P_{i}=\text{max}\;\{P_{i},\;P_{i}\in ∆\text{Sthr}\}$ (7), then i represents the predicted future state of the current frame slot reservation prevalence.
(7)$$P_{i}=\overset{m}{\underset{k=1}{\sum }}{𝒲}_{k}{P_{i}}^{\left(K\right)},\;i\in E$$

By repeating steps 4 through 8, we can predict the slot reservation condition for the next frame after determining the current frame’s slot reservation frequency and adding it to the original series. The cluster head with the best stable neighbors is chosen as the CH with the lowest weight value $\left({𝑤}_{i}\right)$ based on the calculation of the combined ${𝑤}_{i}$. In contrast to the EWCA, all other CHs in the cluster follow the same steps from step two to step nine. W_i_ is equal to the sum of the weighting factors (wf) in this equation, which is wf1 + wf2 + wf3 + wf4 = 1. Table 1 shows that the group of weighting factors (0.47, 0.24, 0.24, and 0.05) produced the best results in terms of greatest PDR, network throughput, and lowest end-to-end delay. This is due to the weighting variables in this group emphasizing high group mobility (0.47), followed by degree difference (0.24), and distance metrics (0.24) while lowering the impact of cumulative time (0.05) on the (merge cluster head) MCH selection process. If the next CH is chosen based on the maximum number of MCHs and the highest relative direction with the shortest remaining distance, the selected road segment will have strong connectivity, increasing the packet delivery ratio (PDR) and decreasing the MAC delay. On the one hand, if the protocol’s generated delay is based on a high remaining cumulative time with less attention paid to the quantity of MCH, the protocol’s generated delay will be high, especially in low traffic density cases; on the other hand, the packet delivery ratio (PDR) will be unaffected because the access method is the same as IEEE 802.11p RTS/CTS. Finally, if the relative distance metric is minimal, the protocol delay will improve because of a reduction in the time it takes the vehicles to transmit the packet until it arrives at its destination.

### 3.2. Periodic Access and CH Connectivity Level

The Merge Cluster Head Selection Algorithm (MCHS) utilizing a Stable Weighted Clustering Algorithm is a theoretical model that employs a vehicle weight value for merge cluster head (MCH) selection during the window phase of the cluster merging process (SWC).

The several metrics analyzed for the MCH election process are listed in this section. These metrics include information about the mobility of each cluster head (CH), such as movement direction, road ID, CH mean velocity, CH connectivity level, and cluster head mean distance from its CH neighbors. A CH finds its neighbors by sending out periodic transmissions with mobility information. A CH’s movement direction and a centralized cluster’s total weight value should only be detected by any surrounding CH before it can receive and process its neighbors’ broadcast message. These metrics are utilized to establish a cluster head’s suitability to become a merging cluster head (MCH) since they ensure a CH’s preparedness.
(8)T=1+p (K−1)

For a successful access probability P, the period T, in Figure 4 and Figure 5, to occupy a time slot by a node as it transmits in a frame is given in Equation (8).
(9)$$T=\left[\frac{\begin{array}{c} 1+P_{\mathrm{ij}}\;\left(K-1\right) \end{array}}{CH_{i}}*\left({\mathrm{Wp}}_{h}-{\mathrm{Wp}}_{1}\right)+{\mathrm{Wp}}_{1}\right]$$

The greatest window period ${\mathrm{Wp}}_{h}$ and the lowest window period ${\mathrm{Wp}}_{1}$ contention values are employed based on the total number of cluster head (${\mathrm{CH}}_{i}$). The duration T required for a node to successfully occupy a time slot when it transmits in a frame is described in Equation (9), similar to [17].

### 3.3. Merging Channel Access Mechanism

During cluster merging, the four channel access modes in the CH are super-frames that can easily adapt to new traffic levels while maintaining stable transmissions. The CH periodically polls clusters for traffic. If the traffic value of two consecutive rounds is significantly different from the initial traffic value, the current access mode for the traffic level is used. The CH gathers data about traffic concentrations and the probability of channel-based merging collision and compares them to the threshold value of various traffic levels during the cluster merging window, as follows: Low traffic level is when the traffic load is lower than the threshold values (Ltv). Light traffic level is when the traffic load is lower than the threshold values (Stv) and higher than threshold (Ltv). Medium traffic level is when the traffic load is lower than the threshold values (Htv) and higher than threshold (Stv). High traffic level is when the traffic load is higher than the threshold values (Htv) as illustrated in Figure 5b.

The ECMA access modes between the CHs in a merging cluster are shown in Figure 5b. Random access is used when the traffic is light. The access technique is the same as that of the IEEE 802.11p RTS/CTS. Only the CHs that need to send a packet to other CHs do so by sending the RTS. The CTS packet indicates that the CHs were successfully accessed. On-demand access is used during low traffic. The access of the cluster head is determined dynamically by its message demands and stated in the RTS packet as related to random access.

The clustering time slot reservation access is implemented at a medium traffic level. Only CHs in the same group can reserve and compete for a time slot. Otherwise, they can compete only in the next frame. CHs 1 and 2 did not compete for the same access time slot, whereas nodes 3 and 4 did. CHs 1 and 2 must wait for the next frame. Polling access occurs in high-traffic level when each cluster head receives a CH polling. If a cluster head needs to send data, the other cluster head prepares a time slot. The polling cycle duration was increased to allow for data transmission from the cluster head. After polling the CH1 for readiness, the cluster head also polls CH2 for readiness, and so on, until the cluster head discovers a CH within its transmission range that has data to transfer, at which point the cluster head initiates a cluster merging.

Cluster merging happens as follows: as the cluster merges, two CHs in the transmission range of each other tend to exchange information and reconfigure the CH with the lowest suitability weight value (*W*_i_) to become the MCH. In Algorithm 1, the other CH transforms into Gw and continues attached to the CMs beyond the MCH’s transmission range until all of the CMs are within the MCH’s transmission range and Gw becomes a CM. The CM then enters the cluster and, alongside the other cluster members, decides to join the cluster, be assigned a time slot, and acquires a new CMID. As a result, the rates of re-clustering and merging collisions are reduced, as well as the hidden terminal problems.
**Algorithms 1:** MCH selection in a merged cluster
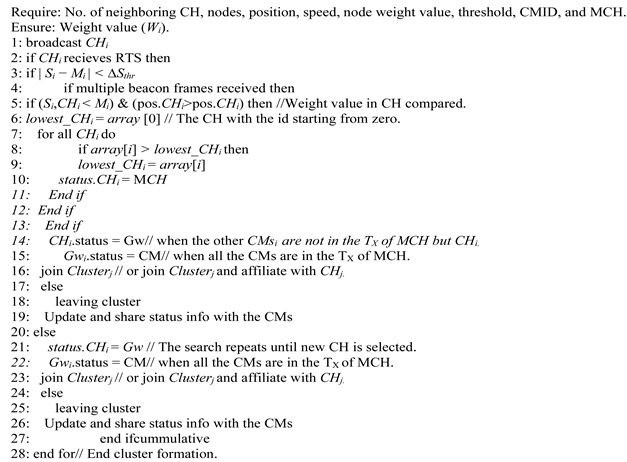


## 4. Performance Evaluation

The simulation results were compared side by side to determine the effectiveness of the proposed algorithm in ensuring an effective cluster merging. We evaluated the performance of the ECMA protocol with the weight-model MCH selection algorithm for efficient transmission of access data packets from the CH and allocate the reserved slot to the CMs to reduce the impact of merging collisions during cluster merging at the simulation phase. The performance metrics for the method are network throughput, end-to-end delay, and access transmission probability.

i.Average network throughput—the average number of data packets successfully transmitted to neighboring CMs within a unit time is known as the average network throughput.ii.The end-to-end delay—the time required for a data packet transmitted and successfully received by neighboring nodes.iii.Successful access transmission probability—defined as the ratio of the number of data packets successfully transmitted in the network to the total number of data packets effectively transmitted.

### Simulation and Parameters

In the simulation, SUMO, NS-2, and MATLAB are used. SUMO is a program that generates road status files by simulating traffic. We used NS2 to embed information about the state of the highway, then evaluated the NS2 to obtain data. We used MATLAB to evaluate data in order to obtain the most important performance indicator information. Wave module is used to communicate across DSRC channels, which are defined by the IEEE802.11p MAC and PHY layer standards. The simulation parameters are listed in Table 1.

## 5. Discussion

### 5.1. Access Delay Time-Slot Probability

Figure 6 shows the probability of access delay theoretical structure based on a weighted Markov chain model: First, as the number of nodes grows, channel access becomes restricted, resulting in access collision.

For example, when the node adapts to different access mechanisms on frames 1 to 4 at *i*th slot 10, ECMA provides 19%, access collisions prevention, while EWCA provides 68.8%. When the transition flow was at *k* and *i*th = 15, the ECMA protocol increased by 15%, whereas the EWCA protocol increased by 82.24%. When two or more clusters merge, the unified weighted cluster network deals with hidden terminal problems and secures all CMs to their CH. The MCH elections and the new merged cluster have a special resistance to merging collisions owing to the transition pace.

### 5.2. Cluster Head Lifetime and Its Influence on Merge Window

The influence of CH duration and the merging window (Mw) is predictable based on the above model and analyses by the simulation process. The MCH selection algorithm with the aperiodic window period also increases the speed of the transition process and generates a new stable merge cluster. This gives the novel ECMA protocol a better performance, even as vehicle densities in a different merge scenario change. Figure 7 attests to the fact that in as much as the density of the vehicles increases from scenarios ‘*a* to *d*’ where the average CH duration during the merging window is low from ‘*a*’ and have a slight rise increases in ‘*b*’, ‘*c*’ and ‘*d*’ scenarios. However, the MCH in the ECMA protocol stays longer than the MCH in the EWCA and VeMAC. Based on the above model and simulation results, the impact of the CH duration and the merging window (Mw) can be predicted. The MCH selection algorithm, combined with the aperiodic window period, speeds up the transfer process and results in a new, stable merge cluster.

### 5.3. Cluster Member Disconnection Frequency and It Influence on Merge Window

When other CHs are relieved of their leadership and become ordinary CMs, the rate of cluster members disconnecting from the network during the cluster merger process is affected. In this case, the CMs must give up their current time slot and request a new one from the new leadership (MCH). CMs that have been separated from their CH during the Mw process may either join the new MCH or leave to form or join another cluster. Figure 8 shows how the CH relinquishes leadership to become the gateway (Gw) node in the ECMA protocol, which continues to connect and link the CMs that are not within the transmission range of the new MCH. In addition, the remaining CMs which are either within the transmission range of the new MCH may cling to the Gate way node or join a new cluster. Second, even though the clusters’ CMs are all within the transmission range of one another, each cluster’s centralized system ensures that each CM is only connected to its own CH. In contrast to EWCA and VeMAC, this function of the ECMA protocol stabilizes and maintains a high-throughput and a timely successful access transmission during the merging window, thereby eliminating the HTP and merging collision.

Even if different cluster members are in the transmission range of each, the centralized network structure (one to all and all to one) using the weighted Markov chain model, where each cluster contains its total weight value, which serves as the cluster ID, significantly connects each CM to its CH. This technique effectively solves the hidden terminal problems, thus eliminating merging collisions. Figure 9 shows that ECMA outperforms VeMAC. Figure 10 demonstrates how the weight-based algorithm is used to achieve a quick transition during the merging process. The construction of a centralized cluster in a vehicle ad hoc network (VANET), where all nodes are one-hop nodes, and aperiodic MCH selection is based on the window period probability. In multi-channel access, the drift in transition dependent on the access mechanism preserves its unique time slot even as this set of nodes transitions from one state to the next. The ECMA protocol outperformed the VeMAC protocol in terms of the transfer speed, resulting in a shorter end-to-end delay. The CHs selects the best candidate to be the MCH in a combined cluster. The other CHs became CMs as their CMs within the transmission range released their time slots and received a new time slot from the new MCH. For a while, the CMs outside of the new CH’s transmission range will stick to their previous CH, which has now switched to Gw Node, until all of the remaining CMs are within the MCH’s transmission range, at which point the Gw is converted to CM. When comparing the ECMA and VeMAC in terms of successful access transmission probability during average velocity, the ECMA protocol in Figure 11 performs better.

## 6. Conclusions

In this study, we proposed an enhanced cluster-based multi-access channel protocol (ECMA) for high-throughput and effective access channel transmissions while minimizing access delay and avoiding collisions during cluster merging. We created a merge cluster head selection (MCHS) algorithm that eliminates merging collision and hidden terminal problems, as well as the selection of the best match MCH in the merged cluster when two or more clusters merge. When multiple sets of vehicles collide or when two or more clusters merge, MCHS algorithm resolves hidden terminal concerns and connects all CMs to their CH. The MCH elections and the new merged cluster are particularly resistant to merging collisions owing to the rapid transition from one state to the other. In high-speed merge, a weighted Markov chain model is used to describe the transformation operation within a cluster. The application of a weighted Markov chain model represents the transformation operation within a cluster and distinguishes it from other clusters based on the weighted value. In addition, the weighted Markov chain mathematical model enhances accuracy while decreasing ECMA channel data access transmission delay with unmatched transition speed in timeslot and state-slot during the clusters’ merger window. These speeds up transitions while avoiding hidden terminal issues and merging collisions during the window period (T). Extensive simulation data were supplied to demonstrate the effectiveness of the proposed strategy. In summary, this work gives a detailed discussion of the basic ECMA protocol modeling and the MCHS algorithm, as well as a thorough analysis of their technology. The discussion concluded that cluster member disconnection frequency is minimal, as well as a longer cluster head lifetime and a positive influence on merge window. Therefore, when ECMA is compared to EWCA and VeMAC, the weighted MCHS algorithm and weighted Markov chain yields a distinct output in terms of average network throughput, end-to-end delay, and efficient access transmission probability by 64.20%–69.49%, avoiding HTP and eliminating merging collisions. In the future, the proposed MCHS algorithm will be assessed in traffic scenarios involving vehicles driving in opposite directions with heterogeneous radio access in order to facilitate information transmission between cluster heads.

## Figures and Tables

**Figure 1 sensors-22-04861-f001:**
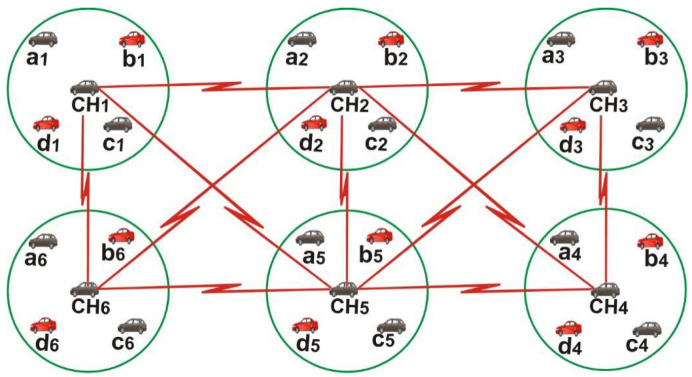
Merging cluster structure.

**Figure 2 sensors-22-04861-f002:**
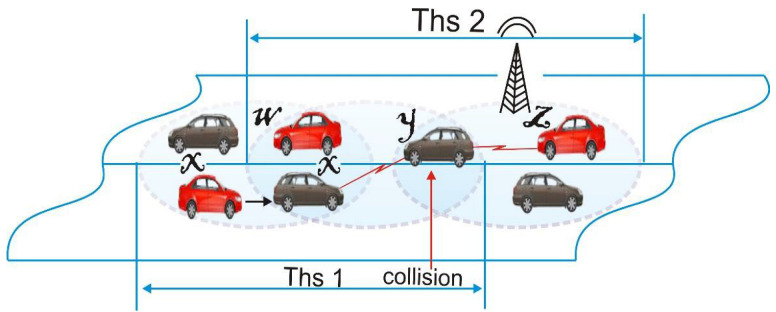
Merge collision due to node mobility [20]. Note: THSO is the ratio of a THS’s necessary time slots to the total number of time slots available for that THS.

**Figure 3 sensors-22-04861-f003:**
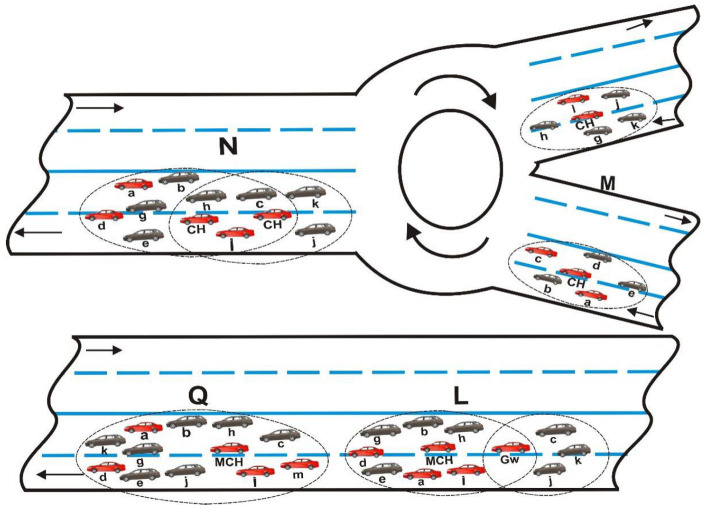
Illustrations of cluster merging phases (from M –> N –> L –> Q).

**Figure 4 sensors-22-04861-f004:**
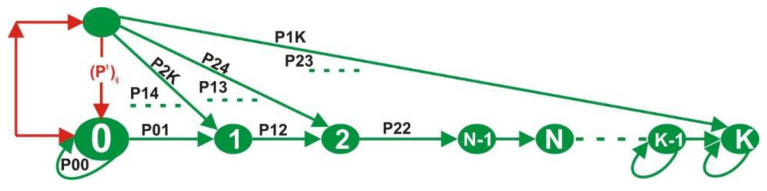
The transition process of *X*_n_.

**Figure 5 sensors-22-04861-f005:**
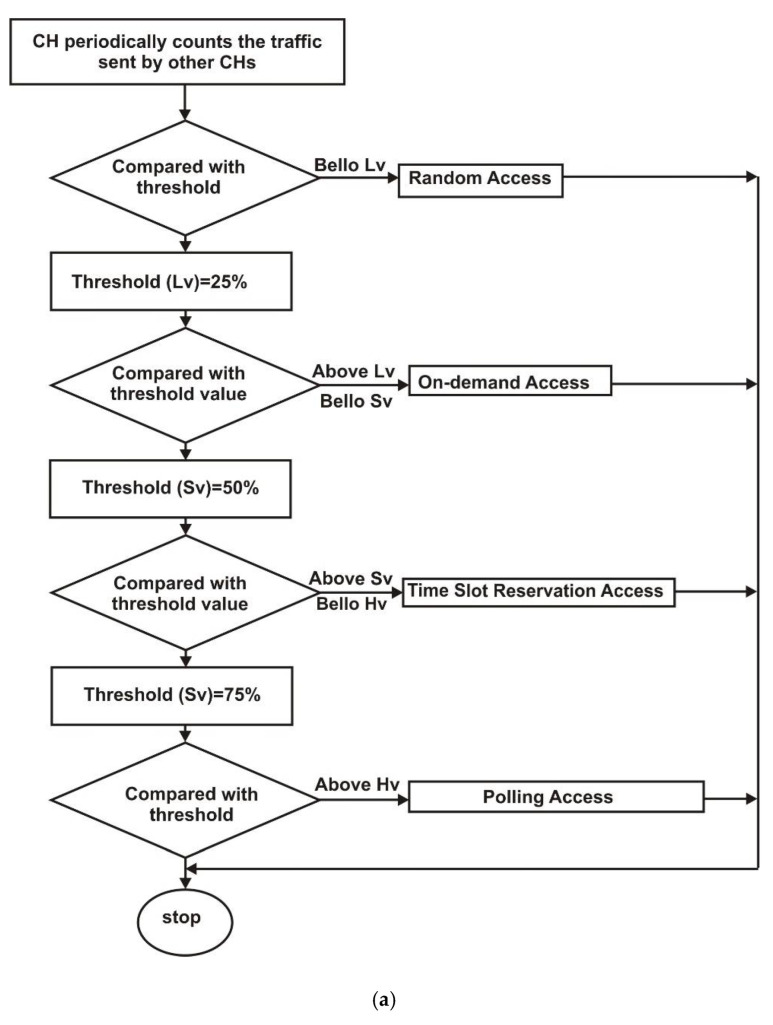
(**a**): Diagram of ECMA protocol. (**b**): Merge cluster head channel access.

**Figure 6 sensors-22-04861-f006:**
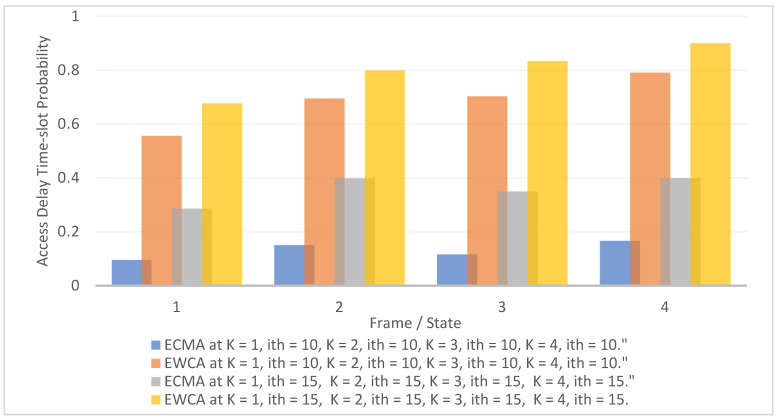
Access delay at various state.

**Figure 7 sensors-22-04861-f007:**
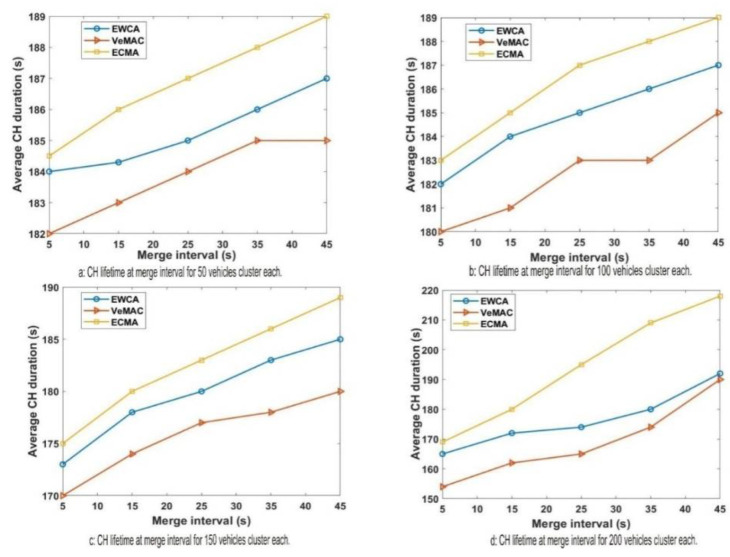
CH lifetime and it influence on Mw.

**Figure 8 sensors-22-04861-f008:**
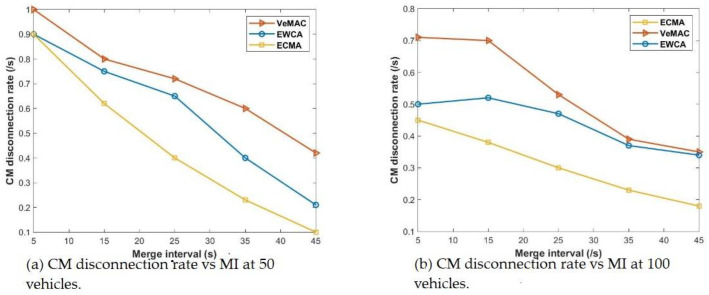
CM disconnection frequency and it influence on Mw.

**Figure 9 sensors-22-04861-f009:**
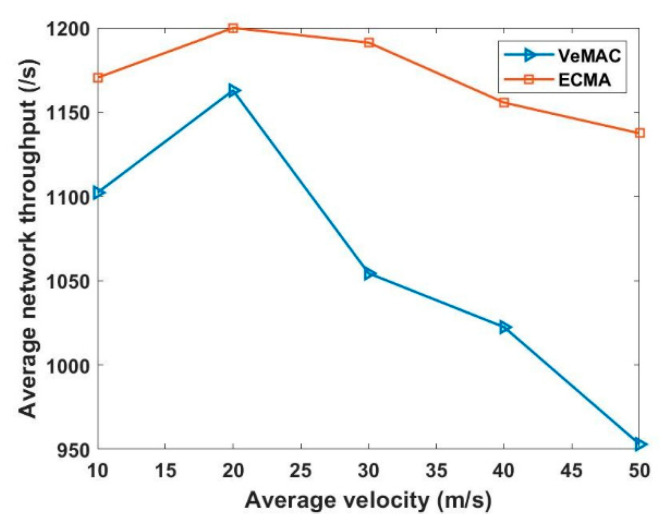
Average network throughput versus average velocity.

**Figure 10 sensors-22-04861-f010:**
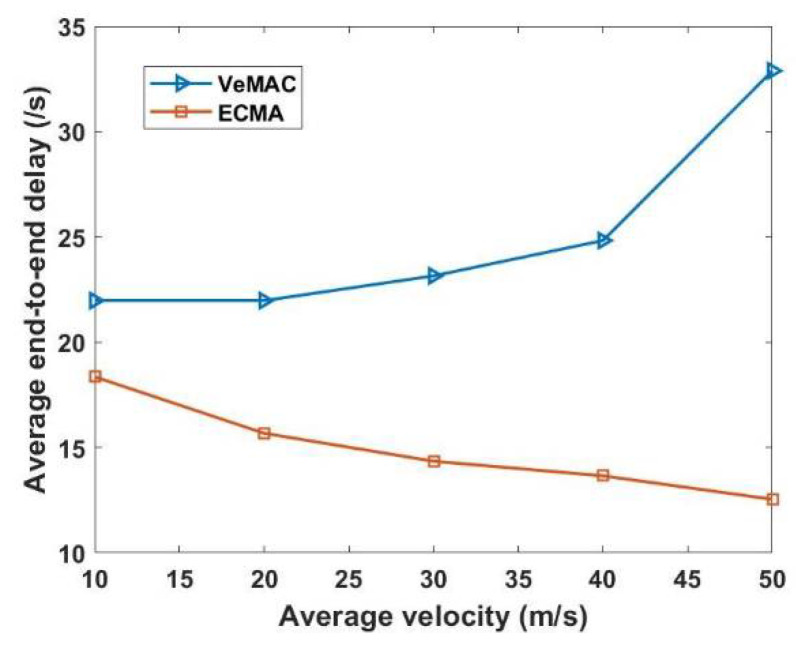
Average end-to-end delay versus average velocity.

**Figure 11 sensors-22-04861-f011:**
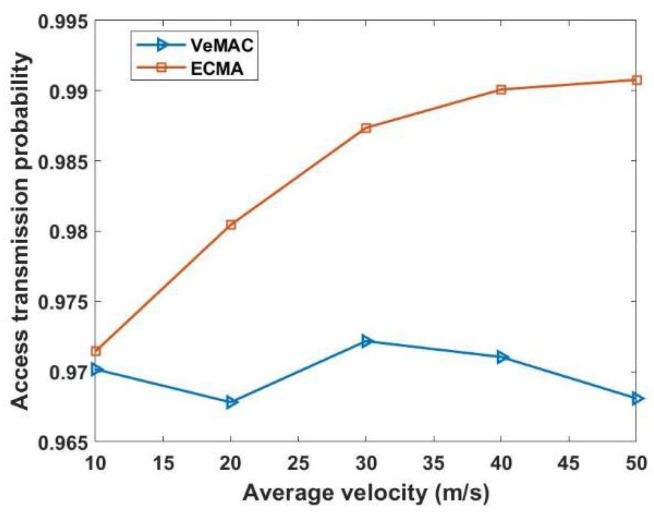
Successful access transmission probability versus average velocity.

**Table 1 sensors-22-04861-t001:** Simulation Parameters.

**Parameter**	**Value**	**Parameter**	**Value**
DSRC channel frequency	5.9 GHz	DSRC channel bandwidth	10 MHz
MAC/PHY	WAVE/IEEE 802.11p	Mean deviation	0-V_max_
Simulation time	1000 s	Vehicle densities	50, 100, 150, 200
Merge window (Mw)	5, 15, 25, 35, 45	Weight factor level	0.47, 0.24, 0.24, 0.05
Radios r	500 m	Region’s size	1000 × 1000
Data rate	100 Mbps	Packet arriving rate	25 Packets/s

## Data Availability

Data available on request due to restrictions eg privacy or ethical.

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
