# Peer review of "A Reliable Merging Link Scheme Using Weighted Markov Chain Model in Vehicular Ad Hoc Networks"

_sensors, 2022, doi:10.3390/s22134861_

Round 1

Reviewer 1 Report

The author properly solved the collision method used in the ad-hoc network, and it was interesting to solve the problem by connecting different access methods in Fig 5. It contains content related to the resolution of the collision method, which is one of the VANET problems. A general revision should be made to make the diagrams and formulas clearer. Performance improvement is included in the conclusion part, but it should include how it can be applied to the part mentioned in the introduction and how future research will proceed. The contents of the introduction are too long, so the contents should be clearly organized focusing on motivation and contribution. Context makes sense, but does not contain a definition for Xn. Variables used in formulas are not clear. (For example, the variable l in Equation 5) The ambiguity in the part shown in Figure 4 should be compensated.

Author Response

Response to Reviewer 1 Comments

Point 1: The author properly solved the collision method used in the ad-hoc network, and it was interesting to solve the problem by connecting different access methods in Fig 5. It contains content related to the resolution of the collision method, which is one of the VANET problems.

A general revision should be made to make the diagrams and formulas clearer.

Response 1: The diagram is clearly defined, explained and corrected in the pages 8 – 9.

The following formulas are explained and corrected: The general time-slot transition step is given by  for any slot time t. The Markov chain nodes X0, X1,…,Xn with a slot time reservation state, S = (1, 2, 3,..... n), (figure 4) where the Transition Matrix  element is defined as: (pages 6 - 7).

Point 2: Performance improvement is included in the conclusion part, but it should include how it can be applied to the part mentioned in the introduction and how future research will proceed.

Response 2: In the future, the proposed MCHS algorithm will be assessed in traffic scenarios involving vehicles driving in opposing directions with heterogeneous radio access in order to facilitate information transmission between cluster heads.

Point 3: The contents of the introduction are too long, so the contents should be clearly organized focusing on motivation and contribution.

Response 3: Corrections taken.

Point 4: Context makes sense, but does not contain a definition for Xn. Variables used in formulas are not clear. (For example, the variable l in Equation 5) The ambiguity in the part shown in Figure 4 should be compensated.

Response 4: The diagram is clearly defined, explained and corrected in the pages 8 – 9.

Reviewer 2 Report

Generally speaking, the paper is not readable at all. The authors should refer to papers from good quality IEEE journals to see how good papers are written. They should get their paper read by a local expert before submission to this journal. 

Some specific observations to improve the paper are as follows: 

  1. Explain the merge collision better by explaining the clustering and slot allocation process in detail. Moreover, argue why one should use the clustering and slot allocation process. There are many competing methods of channel access proposed for vehicular networks but the authors directly start from their narrow area of focus without putting things in right context/perspective. 
  2. The theory of Markov chains is not required to be included in the paper. It is unclear to me exactly what your Markov model is doing and where it is used for clustering/time slot allocation/channel access. 
  3. In general, the different parts of the paper are not well connected and the flow of ideas is jittery. 

Author Response

Response to Reviewer 2 Comments

Point 1: Explain the merge collision better by explaining the clustering and slot allocation process in detail. Moreover, argue why one should use the clustering and slot allocation process. There are many competing methods of channel access proposed for vehicular networks but the authors directly start from their narrow area of focus without putting things in right context/perspective.

Response 1: Corrections taken, in pages 1 – 3 and others.

Point 2: The theory of Markov chains is not required to be included in the paper. It is unclear to me exactly what your Markov model is doing and where it is used for clustering/time slot allocation/channel access.

Response 2: In the vehicular ad hoc approach, the Markov Chain mathematical tool is crucial. While a Markov Chain can be used to predict future occurrences, the current situation determines its importance. Because the stochastic process forecasts the future, channel access transmission speed is unrivaled. V. Nguyen Examines multiple data packet transmission probabilities to predict throughput and latency with accuracy. During the window period (T), the Chain Markov model was employed to quicken transitions while avoiding hidden terminal problems and access conflicts. Furthermore, time-slot uniqueness provides dynamic channel access during slot reservation, reducing access collisions during slot allocations based on numerous access approaches and threshold values that allow super-frames (access mechanism) to transit to one another.

Reference,

Nguyen et al., “A Survey on Adaptive Multi-Channel MAC Protocols in VANETs Using Markov Models,” IEEE Access, vol. 6, pp. 16493–16514, 2018, doi: 10.1109/ACCESS.2018.2814600.

Shafiq, Z., Abbas, R., Zafar, M. H., & Basheri, M. (2019). Analysis and Evaluation of Random Access Transmission for UAV-Assisted Vehicular-to-Infrastructure Communications. IEEE Access, 7, 12427–12440. https://doi.org/10.1109/ACCESS.2019.2892776

Etc.

Point 3: In general, the different parts of the paper are not well connected and the flow of ideas is jittery.

Response 3: The corrections taken.

Reviewer 3 Report

  1. Authors are advised to review the abstract section to better highlight your contributions and methodology for solving the problem. Increase the introductory part. 2. Check the text carefully. Typos and grammar errors were found in the manuscript. Please correct such problems. 3. Increase the quality of the 1-2-3 figures. 4. Increase Bibiography:
    Bibiolagraphic suggestions

Managed Lane as Strategy for Traffic Flow and Safety: A Case Study of Catania Ring Road

Cafiso, S.Di Graziano, A.Giuffrè, T.Pappalardo, G.Severino, A. Sustainability 

Author Response

Response to Reviewer 3 Comments

Point 1: Authors are advised to review the abstract section to better highlight your contributions and methodology for solving the problem. Increase the introductory part.

Response 1: The abstract and the introductory part are clearly defined, explained and corrected in the pages 1 -3.

Point 2: Check the text carefully. Typos and grammar errors were found in the manuscript. Please correct such problems.

Response 2: The corrections taken.

Point 3: 3. Increase the quality of the 1-2-3 figures.

Response 3: The corrections taken in pages 2, 4 and 5.

Point 4: 4. Increase Bibiography: Bibiolagraphic suggestions

Response 4: The corrections taken in pages 15 - 17.

Round 2

Reviewer 1 Report

In the revised paper, a detailed description of the clustering method was written in the background, and the improvements of the paper were well specified by adding specific quantitative indicators to abstract and conclusion.

In the revised paper, the parts that may be confused by the specification of overlapping CHs were distinguished and specified by MCH.

A clear indication of the xn definition presented as review comments and a further description of the probability value were properly reflected.

Hidden terminal problems --> change to hidden terminal occurrence

In the revised paper, the formula and background were well supplemented, but some typos should be corrected, and a clear definition of the words should be made.

Author Response

Response to Reviewer 1 Comments

Point 1: In the revised paper, a detailed description of the clustering method was written in the background, and the improvements of the paper were well specified by adding specific quantitative indicators to abstract and conclusion.

In the revised paper, the parts that may be confused by the specification of overlapping CHs were distinguished and specified by MCH.

A clear indication of the xn definition presented as review comments and a further description of the probability value were properly reflected.

Hidden terminal problems --> change to hidden terminal occurrence

In the revised paper, the formula and background were well supplemented, but some typos should be corrected, and a clear definition of the words should be made.

Response 1: Correctons taken.

Point 1: Are the methods adequately described?

Response 1: The methods include the weighted Markov chain model, which incorporates both the weight value and the Markov chain model, periodic access and the CH connection level, and the merging channel access mechanism. The methodologies were used as a guide to ensure a thorough knowledge. The description / diagram of ECMA protocol based on threshold values in pages 8 & 9.

Reviewer 2 Report

The authors have tried to answer my queries. But there is no real implementation / demonstration. So, it is unclear how useful this work will be in practice.  

Author Response

Response to Reviewer 2 Comments

Point 1: The authors have tried to answer my queries. But there is no real implementation / demonstration. So, it is unclear how useful this work will be in practice. 

Response 1: The cost of deploying and testing VANETs is significant, and it takes a long time. As an alternative to actual implementation, simulation is a valuable and less expensive option. It is necessary to create realistic models in order to obtain good results from VANET simulation, which is a difficult task given the complexities of the VANET infrastructure (e.g., simulators need to model both mobility patterns and communication protocols). Therefore, SUMO, NS-2, and MATLAB were used to implement the simulation approach in this work. SUMO uses traffic simulation to create road status files. We used NS2 to embed files with highway status information, and then analysed the NS2 data. To collect the crucial performance indicator information, we imported the data into MATLAB for analysis. The IEEE802.11p MAC and PHY layer specifications provide DSRC channels, which are used by the wave module(Hadded et al., 2018).

Reference

Hadded, M., Muhlethaler, P., & Laouiti, A. (2018). TDMA Scheduling Strategies for Vehicular Ad Hoc Networks: From a Distributed to a Centralized Approach. 2018 26th International Conference on Software, Telecommunications and Computer Networks, SoftCOM 2018, 164–169. https://doi.org/10.23919/SOFTCOM.2018.8555781

Point 2: Does the introduction provide sufficient background and include all relevant references?

Response 2: Additional and relevant References have been added to the background.

Point 3: Are the methods adequately described?

Response 3: The corrections taken.

Weighted Markov chain model, which incorporates both weight value and Markov chain model, periodic access and CH connection level, and merging channel access mechanism are among the approaches used. The methodologies were used as a guide to ensure a thorough knowledge. The description / diagram of ECMA protocol based on threshold values in pages 8 & 9.

Point 4: Are the results clearly presented?

Response 4: The result clearly presented,

Point 5: Are the conclusions supported by the results?

Response 5: The corrections taken.

The conclusions summaries the entire work and the simulation result in a standard which is updated and the future work added.